# Scenic spot path planning and journey customization based on multilayer hybrid hypernetwork optimization

Chunqiao Song *

College of Tourism, Xinyang Normal University, Xinyang, Henan, China

* songchunqiao2023@163.com

**Data Availability Statement:** All files are available from the repository KTS dataset, https://zenodo.org/records/3381859, DOI: 10.5281/zenodo.1213195.

## Abstract

In the face of increasingly diverse demands from tourists, traditional methods for scenic route planning often struggle to meet these varied needs. To address this challenge and enhance the overall service quality of tourist destinations, as well as to better understand individualized preferences of visitors, this study proposes a novel approach to scenic route planning and itinerary customization based on multi-layered mixed hypernetwork optimization. Firstly, an adaptive multi-route feature extraction method is introduced to capture personalized demands of tourists. Subsequently, a personalized tourist inference method based on a multi-layered mixed network is presented, utilizing the extracted personalized features to infer the true intentions of the tourists. Lastly, we propose a hypernetwork optimized route planning method, incorporating the inference results and personalized features to tailor the optimal touring paths for visitors. The results of our experiments underscore the efficacy of our methodology, attaining an accuracy score of 0.877 and an mAP score of 0.881 and outperforming strong competitors and facilitating the design of optimal paths for tourists.

## 1. Introduction

Amidst the flourishing growth of the tourism industry, the needs of tourists are becoming more and more personalized, and they want to be able to customize their travel paths according to their interests, time and preferences [1, 2]. At the same time, scenic spots also need to improve the service level and occupy a competitive advantage in innovation to improve tourist satisfaction and attract more tourists. Therefore, path planning and itinerary customization of scenic spots has emerged as a crucial research focus within the realm of tourism studies, which involves many aspects, including technical, social and economic levels [3]. As information technology advances swiftly, the widespread incorporation of intelligent and digital technologies has become prevalent in the tourism domain. This integration often involves leveraging Geographic Information System (GIS) and artificial intelligence (AI), big data and other technologies, path planning and itinerary customization of scenic areas can provide tourists with a more intelligent and personalized tourism experience, which is helpful to understand tourists'

**Funding:** The author(s) received no specific funding for this work.

**Competing interests:** The authors have declared that no competing interests exist.

behaviors, preferences and trends [4, 5]. Tourism route planning research involves multiple aspects, including understanding the destination, developing the itinerary theme, designing diversified travel routes, arranging a reasonable schedule, selecting transportation modes, arranging accommodations, planning meals, and organizing activities. By strategically planning the tour path, it is possible to alleviate congestion in specific areas of the scenic spot, optimize resource utilization, and mitigate adverse environmental impacts. The research on path planning and itinerary customization of scenic areas can adapt to the rapidly changing needs of the tourism industry, use advanced technology to provide tourists with more intelligent and personalized travel experience, and promote the sustainable development of tourism, which has very great application value.

The endeavor of researching path planning and itinerary customization for scenic spots encompasses numerous challenges encompassing diverse aspects like technology, data, and user behavior [6, 7]. First, path planning of scenic spots needs to consider multiple factors, including tourists' interests, time, preferences, traffic conditions, and scenic capacity, etc., and the complex relationship between these factors needs to be accurately modeled and described [8]. Second, the condition of the scenic spot is always changing, including pedestrian flow, weather, traffic and other factors. Therefore, path planning and itinerary customization need to be real-time and dynamic, and the recommended path can be adjusted at any time to adapt to the actual situation [9]. Then, the personalized needs of passengers vary greatly, and how to mine the real needs of users from a large number of user data to provide more accurate customized services is a challenge [10]. Finally, the path planning and itinerary customization necessitates the development of intricate algorithms [11], which must take into account a myriad of variables and constraints. These algorithms need to be able to provide high quality personality while ensuring computational efficiency.

Focusing on the above difficulties, many researchers have carried out research on path planning of scenic spots. Among them, Dijkstra [12, 13] is the most classical approach to address the dynamic path planning problem. Zhang et al. [14] optimized the Dijkstra algorithm by dividing the scenic spots and narrowing the consideration range of nodes to improve the calculation speed. Li et al. [15] try to address the blindness of the traditional ant colony (AC) in path search by introducing the threshold sorting algorithm to optimize the search path, and finally solved the optimal path planning problem between two points. Shen et al. [16] first optimized the path heuristic function, and finally verified the stability and effectiveness of the method by simulation experiments. Gao et al. [17] used the back simulated annealing algorithm to choose the ant walking path, and solve the problem of scenic spot planning. Zhang et al. [18] first improved the AC algorithm, and then used the optimization algorithm to carry out the shortest path planning in the multi-scenic spot problem, which verified the feasibility of the optimization algorithm. Zhou et al. [19] introduced a new intelligent travel planning, which provides tourists with tour routes according to their specific needs by establishing a data information database of scenic spots. Li et al. [20] introduced a approach by various constraints in the framework of Geometric Algebra (GA). The algorithm is verified by simulation with the goal of searching the optimal travel path with different constraint combinations in the road network. Chen et al. [21] combined regional path planning and improved genetic algorithm to formulate specific tour paths for tourists.

However, despite achieving certain success in scenic route planning and itinerary customization, these methods often overlook the personalized needs of tourists. The lack of personalized planning approaches makes it difficult to fully consider the tourists' diverse requirements. This may lead to tourists feeling uncomfortable or disappointed during their trips, and it also fails to fully leverage the unique features and advantages of the scenic spot, thereby reducing the overall quality of the tourism experience. To better meet the comprehensive needs of scenic

route planning and itinerary customization, we need to further explore more intelligent methods. These methods should comprehensively consider factors such as tourists' personalized needs, the geographical and cultural characteristics of the scenic spot, as well as real-time environmental conditions, to create itineraries that not only meet tourists' expectations but also fully showcase the unique features of the scenic spot. Therefore, this paper proposes a path planning and itinerary customization of scenic spots based on multi-layer hybrid hyper-network optimization, so as to build an intelligent travel route recommendation and design system and improve the reception capacity of scenic spots.

## 2. Related works

Scenic spot path planning refers to providing effective and reasonable navigation and path recommendation for tourists in a scenic spot, so that they can better visit the scenic spot and reach their required destination. This involves information collection, data processing, algorithm design and other aspects of the work, including Geographic Information System (GIS) positioning technology, path planning algorithm design, real-time data update and monitoring technology.

The current state of path planning research exhibits notable progress across various methodologies. Gupta et al. [22, 23] pioneered the integration of EMT into traditional evolutionary computation, introducing a multi-factor evolutionary algorithm. SyarifA et al. [24] introduced a fuzzy shortest path model, utilizing fuzzy numbers for evaluation and screening, and applied genetic algorithms for optimization [25]. Wang et al. [26] enhanced genetic algorithms with a large-scale neighborhood search for optimal path solutions, showing superior performance compared to conventional genetic algorithms. Li et al. [27] proposed a novel genetic algorithm using continuous adjacent grid and unequal length chromosome coding for efficient robot obstacle avoidance in discrete grid environments. Fu et al. [28] designed a hybrid genetic algorithm integrating A$^*$ algorithm, demonstrating scalability and efficiency in route optimization problems. Jiang et al. [29] developed a method for generating lane-level maps from sensor data, contributing to improved navigation systems. Ren et al. [30] mapped the search space into a deformable space, utilizing heuristic search for path planning, albeit with high computational complexity. Gonzalez et al. [31] employed a multi-resolution state lattice to enhance planning optimality and speed in trajectory search, particularly effective in complex global maps. Overall, these studies showcase advancements in path planning methodologies, from evolutionary algorithms to fuzzy logic and heuristic search, each contributing to improved efficiency and performance in various path planning scenarios.

In addition, Deep learning technology has significantly advanced path planning research, as evidenced by several notable contributions. Wang et al. [32] integrated deep learning with path planning techniques, employing a convolutional neural network (CNN) to guide the sampling procedure of the RRT$^*$ algorithm, resulting in improved path planning solutions. Q-Jeong et al. [33] utilized the triangle inequality to reduce path nodes in Q-RRT$^*$, thereby decreasing path cost and accelerating convergence, although the sampling procedure remains random. Shiri et al. [34] introduced the oHJB algorithm, an online control algorithm for remote Unmanned Aerial Vehicles (UAVs) assisted by a neural network, enhancing UAV navigation capabilities. Tan et al. [35] proposed an AC-based neural network mission planning technique for heterogeneous multi-UAV systems, leveraging AC optimization to maximize benefits and minimize tasks. Yuan et al. [36] formulated UAV flight path tasks and employed CNN for object detection, coupled with optimization to identify optimal flight paths, enhancing navigation efficiency. Kim et al. [37] introduced a path planning approach utilizing a grid graph and optimization index to obtain a weighted network graph, simplifying route planning

by transforming it into a minimum weight problem. Jabbar et al. [38] proposed a weighted network graph based on polygons, automatically generating shortest routes through an improved Dijkstra algorithm. However, this optimization method may generate multiple alternative routes and exhibit inefficiencies. These advancements underscore the integration of deep learning techniques into path planning, leading to more efficient and effective navigation solutions across various domains.

Although there are numerous algorithms available to enhance path planning, when applied to practical multi-task scenarios, their iterative optimization processes often only handle a single path planning task. This limitation is particularly evident in multi-task environments, which typically involve multiple target points, dynamic obstacles, and real-time map information. Most existing path planning algorithms are based on the single-task assumption, meaning they tackle only one path planning task from start to finish at a time. However, in complex multi-task scenarios, multiple path planning tasks need to be considered simultaneously, and these tasks may have mutual interferences and constraints. Therefore, this paper mainly explores how to effectively represent and handle the complex constraints and interferences in multi-task environments.

## 3. Methods

To improve the scenic spot path planning and realize the personalized customization of tourists 'itinerary, we propose a scenic spot path planning upon multi-layer hybrid hypernetwork optimization. The model aims to meet the comprehensive needs of scenic spots and tourists, including trip diversification, path optimization, multimodal information fusion and other challenges. By introducing advanced multi-layer hybrid hyper-network optimization, the feature extraction method of adaptive multi-path selection, the tourist personalized reasoning method based on multi-layer hybrid network and the path planning method based on hyper-network optimization are proposed in sequence. The efficient matching between tourists and scenic spots is realized.

The schematic representation of our approach is illustrated in Fig 1. First, to capture the behavioral characteristics of tourists more effectively, we have adopted an adaptive multi-path selection method for feature extraction. This method exhibits great flexibility and is able to intelligently select the appropriate feature extraction paths based on different types of information, such as users' historical itineraries and preferences for interesting points, thus more accurately capturing the personalized needs of tourists. Second, we have introduced personalized reasoning for tourists based on a multi-layer hybrid network. By deeply integrating tourist features at multiple levels, we can gain a more comprehensive understanding of their interests and preferences. This method provides more precise personalized information for subsequent

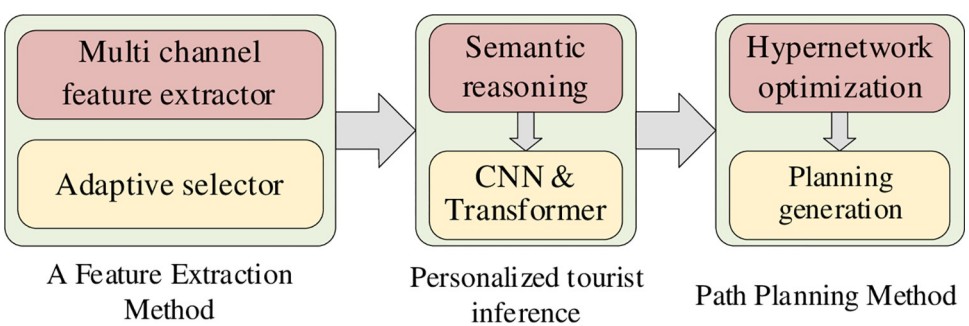

**Fig 1. Scenic area path planning based on multilayer hybrid hypernetwork optimization.**

route planning, contributing to the enhancement of personalized route planning. Finally, we have proposed a route planning method based on hypernetwork optimization, aiming to achieve efficient matching between tourists and scenic areas. By optimizing the hypernetwork structure of route planning, we can find the optimal path solutions while considering multiple factors. This helps to ensure that tourists have a better experience during their visit to the scenic area and improves the overall tourism efficiency. Overall, the route planning method proposed in this article, based on the optimization of a multi-layer hybrid hypernetwork, provides an innovative solution for improving the efficiency of route planning and the level of tourist experience by fully considering the needs of scenic areas and tourists, as well as introducing advanced technological means.

## 3.1 Feature extraction method of adaptive multi-path selection

Adaptive multi-path selection (AMS) feature extraction is a technique used to capture diverse information and flexibly select feature extraction according to different input types. This method aims to improve the adaptability of feature extraction to more accurately meet the requirements of complex and variable application scenarios.

We use tourists' historical trips, points of interest preferences, and social network activities as the targets of feature extraction, which involve geographical location, time and user behavior. In order to capture the characteristics of different tourists more comprehensively, we design the corresponding feature extraction network for each type of tourist characteristics and optimize it through the pre-training stage. In this way, we obtain multiple specific recognition networks, and each network is responsible for extracting the features of a certain type of tourist characteristic. These specific recognition networks are combined into a multi-channel network, and we input different features of tourists into it to obtain richer feature representation, as shown in Fig 2. Then, by introducing an adaptive selection module, we are able to judge the feature domain of different features according to the input features, and classify them accordingly. This process enables the system to intelligently adapt to different types of input information, and ensures that the extracted features are more in line with the personalized needs of each feature.

In the pre-training stage, for the feature $D_i$ of type $i$ tourists where $i \in R$ means the number of the entire types, we adopt a CNN model (for example, ResNet) to train the classifier, so that the model can classify $D_i$ and output the corresponding feature representation, denoted by $\varphi_i$. Next, we construct an adaptive multi-path selection feature extraction method. For a given instance $x_n$, we feed it into all the learned backbone networks simultaneously while keeping the network parameters frozen to be used for target feature extraction, resulting in a set of features. Each feature is associated with a certain tourist feature. Finally, we feed these features

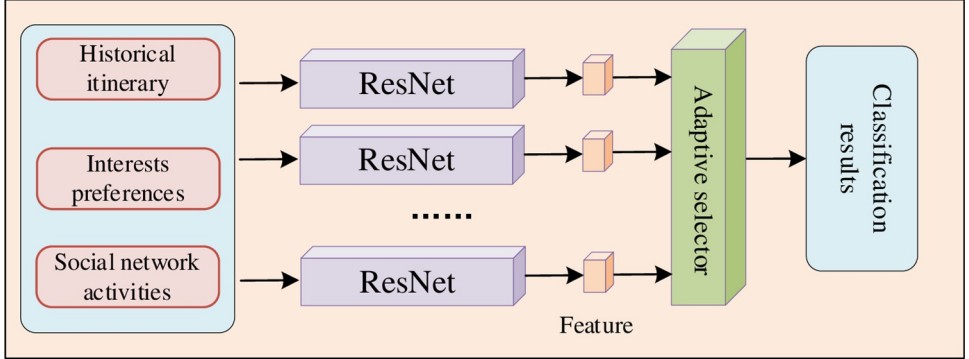

**Fig 2. Adaptive multi-channel feature selection.**

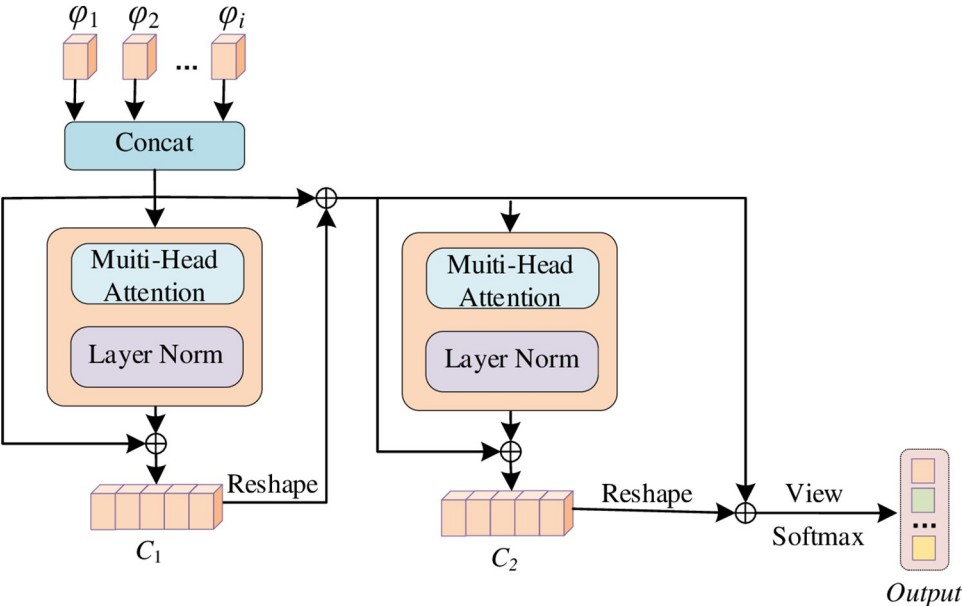

**Fig 3. Adaptive selector.**

into the adaptive selection module, as shown in Fig 3, whose expression is as follows:

$$\begin{cases} \partial = Concat(\varphi_1, \varphi_2, \ldots, \varphi_i) \\ C_1 = \partial + LN(MHA(\partial)) \\ C_2 = \partial + LN(MHA(Re(C_1 + \partial))) \\ F = Softmax(view(C_1 + \partial + C_2)) \end{cases} \quad (1)$$

According to the input features, the adaptive selection module can intelligently determine the different tourist characteristics to which each feature belongs and classify them accordingly. By analyzing and processing the input features, it can accurately determine which kind of tourist feature each feature belongs to. Since each feature is associated with a certain tourist feature, the adaptive selection module can make personalized classification according to the attribution of the feature. This helps to understand tourists' interests, preferences and other personalized information more accurately. The module has certain flexibility and adaptability, which can adapt to different types of input features, making the system more universal. This adaptivity allows the system to respond flexibly and intelligently when faced with visitor data with different characteristics. If the input instance contains multiple tourist characteristics, the adaptive selection module can integrate these characteristics to provide the system with more comprehensive and integrated information. This helps to better meet the comprehensive needs in tasks such as scenic spot path planning. The adaptive selection module plays a key role in the whole system. Through the intelligent judgment of the input features, it provides targeted feature information for subsequent personalized reasoning and path planning, so as to improve the accuracy and effect of the system in different tourist characteristics scenarios.

### 3.2 Personalized reasoning of tourists based on multi-layer hybrid network

Multi-layer hybrid network-based Tourist Personalized Reasoning (MHN) is a method to deeply understand tourists' interests and preferences and provide personalized services for

them. This method uses a multi-layer hybrid network structure to deeply fuse the features of different levels. Each network layer is responsible for capturing information at different levels of abstraction. For example, the lower layer may focus on basic geographic location information, while the higher layer may focus on more abstract behavior patterns or preferences.

Multi-layer hybrid networks simultaneously process input of multiple visitor characteristics, which can include multi-modal information such as historical trips, point of interest preferences, social network activity, and so on. Each feature is represented by a corresponding level of hybrid network, and finally fused at a high level to obtain a more comprehensive and multi-dimensional representation of tourist characteristics. In each hybrid network layer, the information is deeply integrated to understand the personalized needs of tourists more comprehensively.

Firstly, the Transformer is used for text semantic reasoning to reason about the semantic information of the text. Then, the semantic information is used to guide the CNN and Transformer based model to reason about the tourist features, and the recognition results are obtained. The formula of CNN is as follows:

$$O(i,j) = \sum_m \sum_n I(i+m, j+n) K(m,n) \qquad (2)$$

where $O(i,j)$ denotes the dot at $(i,j)$. $I(i+m,j+n)$ refers to the dot at $(i+m, j+n)$ in the data. $K(m,n)$ denotes the weight in the convolution kernel and $m$, $n$ denote the size of the kernel which are odd numbers. This formula represents that each output element in the convolution operation is the weighted sum of the input elements within the coverage of the convolutional kernel. The convolutional layer is able to extract local features from the data and pass these features to the next layer of the network for further processing. The network structure is shown in Fig 4. Semantic inference using Transformer model. The features are encoded by a pre-trained Transformer model to capture their context information. Then, the decoder is used to gradually generate the complete local features, which guide the subsequent generation process and realize the complete recovery of the inference results. Based on the complete semantic information obtained in the previous step, a combination model of CNN and Transformer is used for feature extraction. In this stage, CNN and Transformers are employed to capture both global and local features within the image, and then the local feature prior information and image features are fused and input into the Transformer decoder for deeper processing and fusion of these features to capture richer context information. Finally, the final inference result is obtained by the classifier.

## 3.3 Path planning method based on hypernetwork optimization

The travel planning is to achieve a personalized route with the minimum travel cost for users, so as to improve the user's travel experience as much as possible. Since the planning of

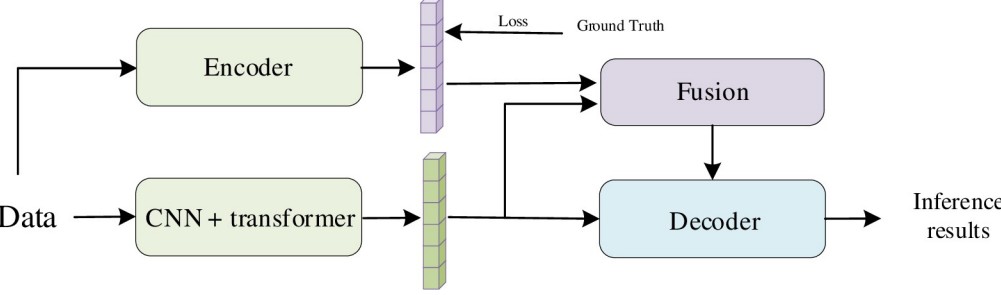

**Fig 4. Personalized reasoning for tourists.**

personalized travel routes needs to satisfy the different users, effectively identifying and obtaining the interest spots selected by users is the key to successfully complete the travel route planning. Therefore, by the features from the above methods and the reasoning results of tourist preferences, we propose a route planning approach by hypernetwork optimization (HO).

Firstly, a user interest spot set $S$ is defined, which contains all the interest spots selected by the user in the existing attraction data database, including the starting attraction, the middle attraction and the final attraction. Then, a user's interest scenic spots subset $S$ is defined, which is a subset of the user's interest scenic spots set divided by the constraint condition $\beta$ in a certain space, and each subset represents a stage in the dynamic programming algorithm. The different scenic spots in the user's interest scenic spots subset $s_j$ can be shown as follows:

$$s_j = S[s_i] \tag{3}$$

Therefore, the best path for tourists between scenic spots can be represented by a GCN with topology, and the network is shown in Fig 5.

In order to optimize the path decision-making ability of Graph Convolution Network (GCN), we mathematically model it. The sequence composed of global decisions starting from an initial state $x_1$ is denoted as $p_{1n}(x_1)$ and is shown in the following equation:

$$p_{1n}(x_1) = \{\mathbf{y}_1(x_1), \mathbf{y}_2(x_2), \ldots, \mathbf{y}_n(x_n)\} \tag{4}$$

When the decision sequence in the sub-process starting from state $x_k$ at stage $k$ to the final state can be denoted as $p_{kn}(x_k)$, the same as the above equation. Then the objective function of the optimal decision is denoted by $f_k(x_k)$, which represents the shortest distance between the state attraction $x_k$ and the final attraction personalized by tourists, and its expression is shown in the following equation:

$$f_k(x_k) = \min_{p_{kn} \in P(x_k)} W(x_k, p_{kn}) \tag{5}$$

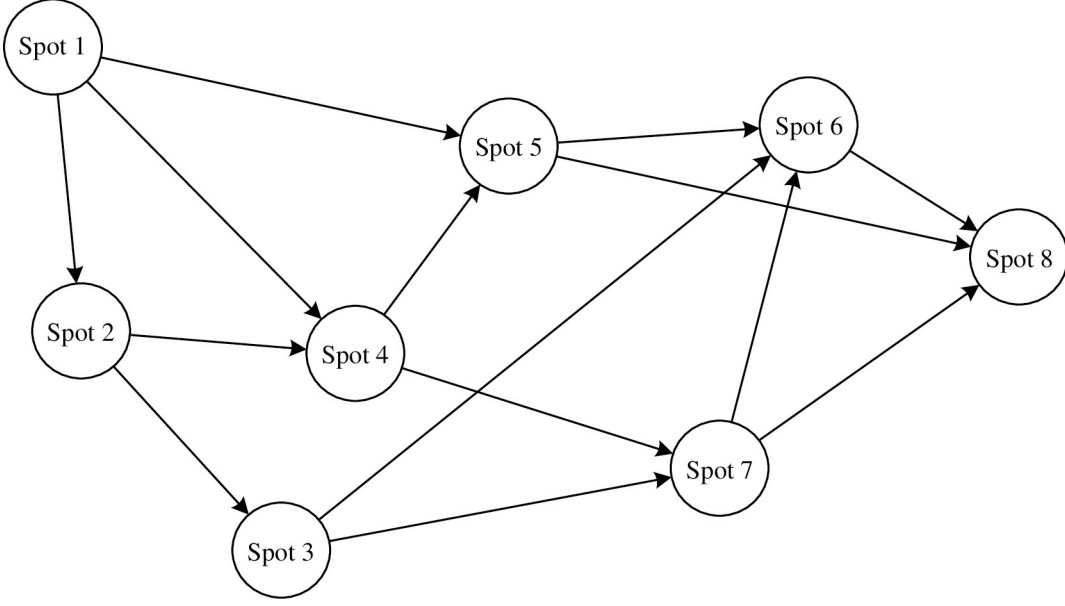

**Fig 5. GCN for scene spots.**

When considering how to plan a tour route to maximize tourist satisfaction, Eq 5 can serve as a crucial mathematical modeling tool to guide our decision-making. In the tourism industry, the design of a route that maximizes tourist satisfaction takes into account factors such as transportation, accommodation, catering, attractions, and tourists' time. By constructing a mathematical model of the above-mentioned scenic spot model and multi-stage decision-making optimization process, this article can effectively achieve route planning for tourist attractions, making the tourism route as scientific and rational as possible, and improving user travel comfort.

## 4. Experiment and analysis

### 4.1 Dataset and implement details

We use the Tourist Spot Dataset (https://zenodo.org/records/3381859, DOI: 10.5281/zenodo.1213195) based on multi-layer hybrid network optimization of path planning of the scenic Spot and custom for testing. The dataset consists of four different modalities—images, text, tags, and likes—which provide a multifaceted perspective of tourist attractions in South Korea. In this dataset, 10 different categories categorize various aspects of the Korean tourism industry, providing a comprehensive representation of various attractions. The data extraction process comes entirely from Instagram, ensuring a real-world description of user-generated content related to South Korean travel destinations. Prior to analysis, the dataset was thoroughly pre-processed, improving the quality and usability of research and exploration. Instagram's careful curation has contributed to a rich and informative dataset that encapsulates the essence of the Korean tourist experience.

We completed the training of the proposed method using a hardware environment configured with i5-12400F and 8*Nvidia RTX4060. The deep learning framework is selected as Mxnet, and the specific training Settings are shown in Table 1. In the model training, we adopted a weight decay term of 0.004. In addition, the initial learning rate is set to $1\times10^{-3}$, the number of epochs is 40, the batch size for training is 32, the decay is set to 0.92, and the gradient descent method is SGD.

In this paper, Accuracy (Acc), Mean Average Precision (mAP), path length (L) and path consumption time (T) are used as the evaluation criteria for cloud computing load detection and identification, and the calculation formula is as follows:

$$Acc = \frac{N(pr)}{N(gt)} \tag{6}$$

$$mAP = \frac{1}{Q}\sum_{i=1}^{Q}\frac{1}{R(i,n)}\sum_{j=1}^{n}\frac{I(i,j) \times R(i,j)}{j} \tag{7}$$

where $pr$ and $gt$ represent the predicted and true values of the model, respectively, and N(.) Represents the function calculation of the statistical value.

**Table 1. Implementation parameters.**

| Parameters | value |
|---|---|
| Initial learning rate | $1\times10^{-3}$ |
| Epoch | 40 |
| Batch-size | 32 |
| Decay | 0.92 |
| Gradient descent method | SGD |

## 4.2 Ablation experiments

On the Tourist Spot Dataset, we conducted ablation experiments of adaptive multi-path selection feature extraction method (AMS), multi-layer hybrid network based personalized tourist inference (MHN), and hypernetwork optimization based path planning method (HO). The results in Fig 6 and Table 2 show that the baseline performance is 0.775 mAP value, 0.782 Acc value, 22 L value, and 234 T value. We employ the CNN as our baseline. Combining the results of attraction recommendations and geographic information data, CNN can assist in planning optimal travel routes. By analyzing information such as the geographic location, traffic conditions, and tourist flow of attractions, CNN can help tourists select the most suitable tour

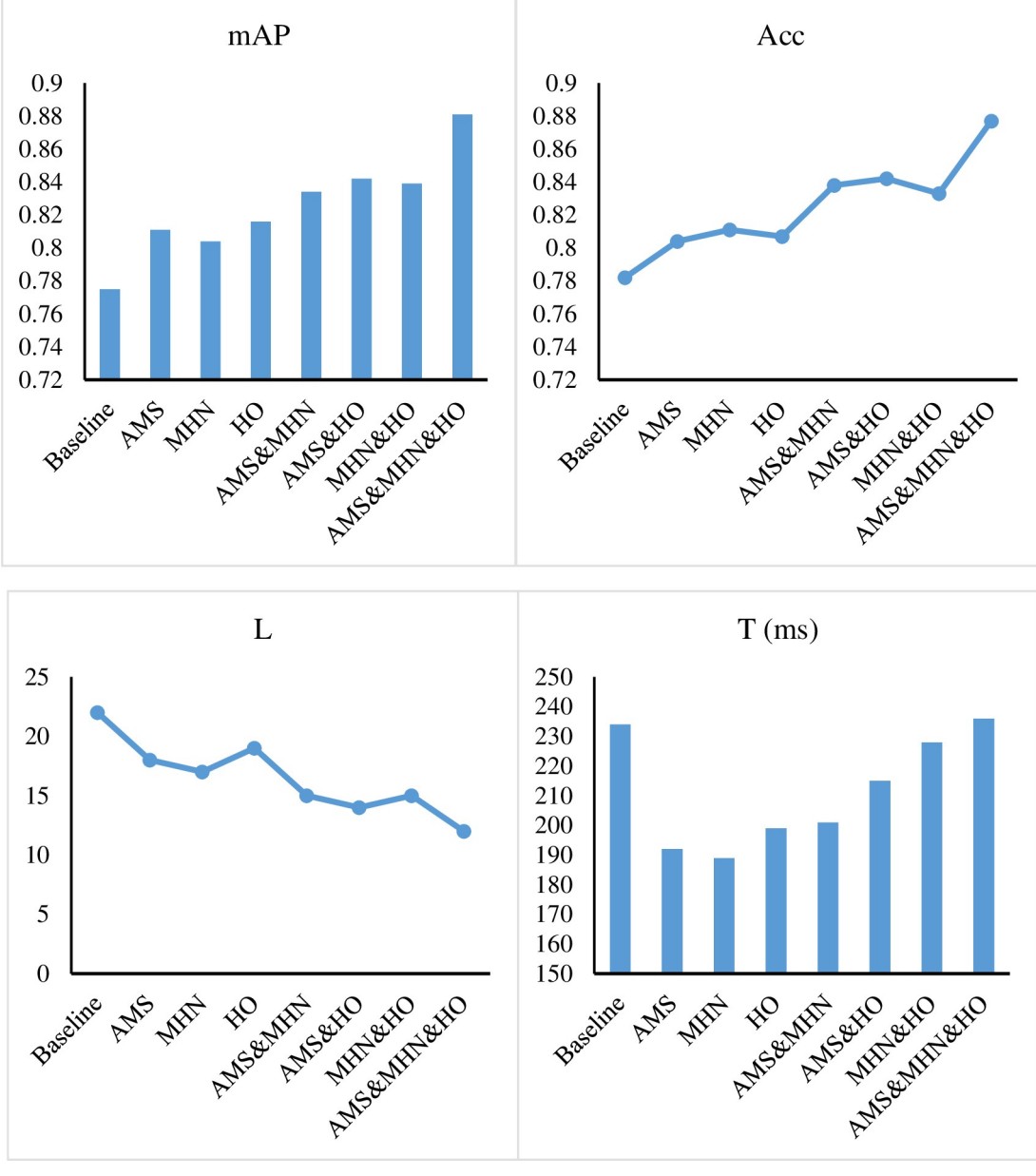

**Fig 6. Ablation experiments results in terms of mAP and Acc.**

**Table 2. Ablation experiments.**

| AMS | MHN | HO | mAP | Acc | L | T (ms) |
|-----|-----|----|----|-----|---|--------|
| Baseline | | | 0.775 | 0.782 | 22 | 234 |
| O | | | 0.811 | 0.804 | 18 | 192 |
| | O | | 0.804 | 0.811 | 17 | 189 |
| | | O | 0.816 | 0.807 | 19 | 199 |
| O | O | | 0.834 | 0.838 | 15 | 201 |
| O | | O | 0.842 | 0.842 | 14 | 215 |
| | O | O | 0.839 | 0.833 | 15 | 228 |
| O | O | O | 0.881 | 0.877 | 12 | 236 |

sequence and transportation mode. Firstly, AMS, MHN and HO modules are embedded separately on the basis of baseline, and the mAP values are 0.811, 0.804 and 0.816, respectively, and the Acc values are 0.804, 0.811 and 0.807, which directly prove the performance of each independent module. Then, the performance of the pairwise combination of the three modules is tested, and the mAP values of 0.834, 0.842 and 0.839 are obtained, and the Acc values of 0.838, 0.842 and 0.833 are obtained, which shows that the pairwise combination of the modules improves about 2% compared with the single module. Finally, when the three modules are used at the same time, the mAP value of 0.881, the Acc value of 0.877, the L value of 12 and the T value of 236 can be obtained. This shows that the combined application of the three modules has a significant improvement in performance.

This experiment proves the combination effect of AMS, MHN and HO on the Tourist Spot Dataset. Combining the three modules, we observe a mAP value of 0.881, which is a significant improvement over the baseline value of 0.775, demonstrating the superiority of this integrated approach. In addition to the improvement in mAP and Acc values, we should also notice the values of L (path length) and T (path time). With all three modules combined, the L value dropped from 22 to 12 and the T value dropped from 234 to 236. This indicates that the application of the integrated module has also achieved positive results in improving the efficiency of path planning. Overall, by comparing the individual application and combination use of different modules, we verify the effectiveness of AMS, MHN, and HO in improving the task performance on the Tourist Spot Dataset. This provides strong support for tourists' personalized reasoning and path planning, and lays a foundation for future related research and practical applications.

## 4.3 Compare our method and other methods

Firstly, we conduct the performance experiments of the adaptive multi-path selection feature extraction method (AMS) on the Tourist Spot Dataset. We selected CNN-FPN [39], Self-Attention [40], Uniformer [41] and LiConvFormer [42] to compare the performance with our method, and the results are shown in Fig 7 and Table 3. We used mAP50, mAP75, mAP95, and mAP to represent the mean average precision (AP) of all classes when the Intersection over Union (IoU) is equal to 0.50, 0.75, 0.95, and 0.50: 0.95, respectively. Compared with other algorithms, AMS achieves the highest AP values in all evaluation metrics, and mAP50, mAP75, mAP95 and mAP achieve 0.811, 0.846, 0.814 and 0.796, respectively, which highlights its robustness and superiority at multiple accuracy levels. Compared with CNN-FPN, our method improves the mAP value of the model by more than 5%. This improvement shows that the adaptive multi-channel selection method proposed in this paper can fully express the information of tourists and improve the accuracy and quality of data. Compared with Self-Attention and Uniformer, our method is in the lead in all indicators. This is attributed to the adaptive multi-path selection mechanism of AMS, which makes the model more adaptive and

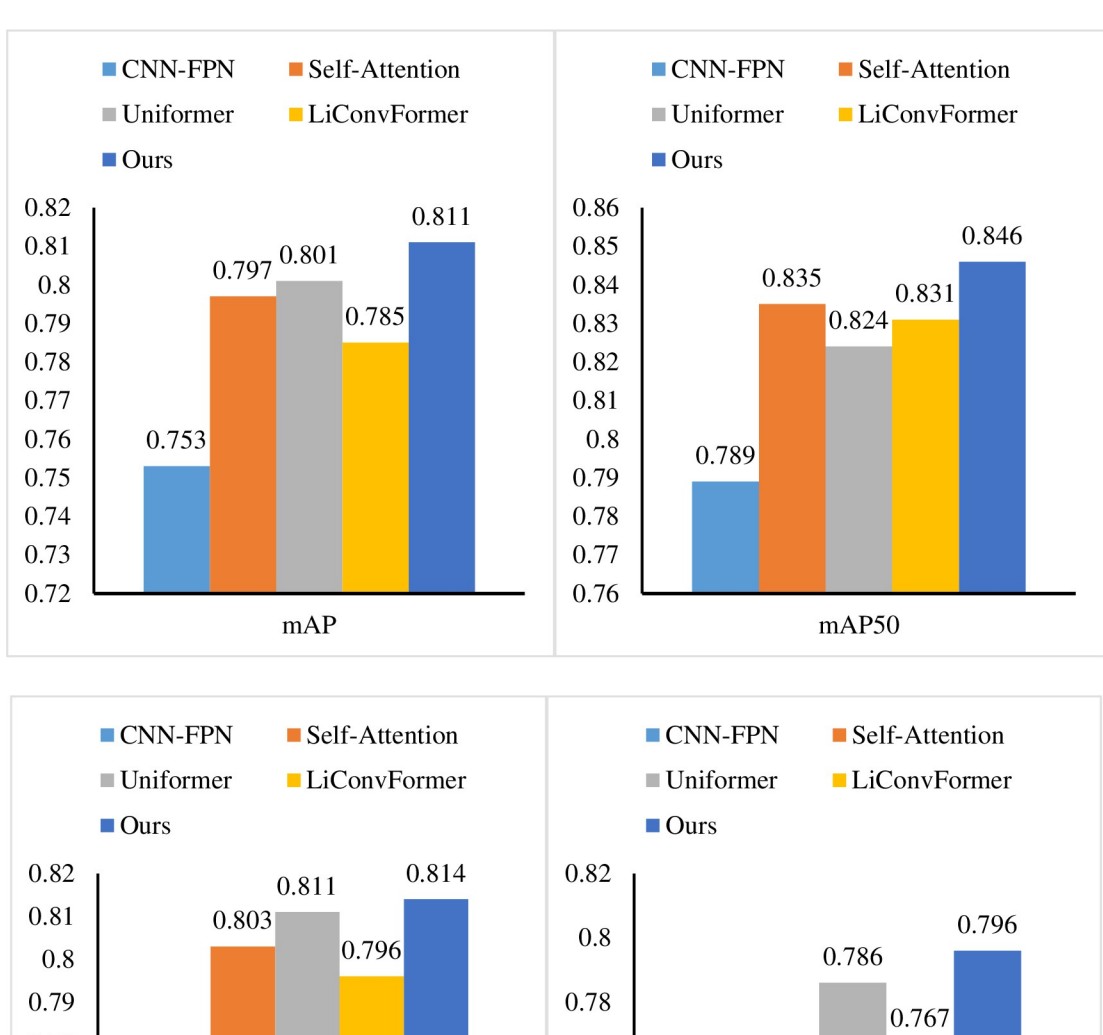

**Fig 7. Comparison AMS with other extraction methods.**

**Table 3. Comparison AMS with other methods.**

| Methods | mAP | mAP50 | mAP75 | mAP95 |
|---|---|---|---|---|
| CNN-FPN | 0.753 | 0.789 | 0.762 | 0.732 |
| Self-Attention | 0.797 | 0.835 | 0.803 | 0.753 |
| Uniformer | 0.801 | 0.824 | 0.811 | 0.786 |
| LiConvFormer | 0.785 | 0.831 | 0.796 | 0.767 |
| Ours | 0.811 | 0.846 | 0.814 | 0.796 |

**Table 4. Compare KPD with other methods.**

| Methods | Acc | GFLOPs | Cost time (ms) |
|---|---|---|---|
| SCN_GNN | 0.831 | 199 | 110 |
| RT-GCN | 0.829 | 211 | 135 |
| MSA-GCN | 0.815 | 204 | 121 |
| Ours | 0.838 | 193 | 102 |

flexible, and can better adapt to different scenarios and data distributions, thereby improving the accuracy. Compared with LiConvFormer, our method has a mAP95 value of 2.9%. This result further proves that AMS can accurately express the characteristics of tourists. In general, through the comprehensive performance experiments on the Tourist Spot Dataset, AMS not only performs well in various evaluation indicators, but also shows obvious superiority over other algorithms, which provides an effective solution for improving the accuracy and quality of tourist information.

We then conduct a more in-depth analysis of the performance of MHN to explore its superiority in personalized reasoning for tourists. As shown in Table 4, from the perspective of Acc value, the score of 0.838 indicates that MHN achieves a significant breakthrough in inference accuracy. This proves that MHN is superior to other models when dealing with complex tourist preferences and behavior patterns. Taking a closer look at Fig 8, we can see that MHN performs well on different precision-computational complexity trade-offs. This balance is achieved by MHN's unique hybrid network structure and weight adjustment mechanism, which enables the model to effectively control the usage of computational resources while maintaining high accuracy. This is crucial for achieving high performance personalized reasoning in resource-constrained practical application scenarios. In terms of computational complexity, MHN shows a satisfactory performance, consuming only 193 GFLOPs. Compared with SCN_GNN [43], RT-GCN [44] and MSA-GCN [45], MHN has higher utilization of computational resources in the inference process, which means that it can complete more inference tasks in the same time, improving the overall efficiency of the system. In addition, in terms of inference time, MHN only needs 132ms, showing excellent real-time performance. Compared with other models, MHN has advantages in response speed and provides better support for immediate decision making in practical application scenarios. In general, compared with SCN_GNN, RT-GCN and MSA-GCN, the results of this experiment clearly show that MHN has significant advantages in tourist personalized reasoning, which not only leads in accuracy, but also performs well in computing resource consumption and reasoning speed. This makes MHN an ideal choice for processing large-scale tourist data sets, and provides more powerful technical support for the personalized service of tourism business.

Finally, we conduct a thorough investigation into the performance of the hyper-network optimization-based path planning method (HO), and we utilize a nonlinear objective function for all comparisons. It can be observed from Fig 9 that HO achieved impressive results in terms of two key performance indicators: Acc and mAP. In terms of Acc, HO scored a remarkable 0.877, demonstrating its excellent performance in path planning accuracy. This indicates that HO is capable of more accurately deriving the optimal solution when dealing with complex path planning scenarios, making the path planning more aligned with actual needs. Similarly, in terms of mAP, HO scored 0.881, surpassing comparison methods such as TriPField [46], CAtNIP [47], and Fields2Cover [48]. This further verifies HO's outstanding performance in path planning diversity and global optimization. HO not only focuses on accuracy but also takes into account the overall quality of path planning, making the generated paths more practical and effective. Overall, the hyper-network optimization-based path planning method, HO,

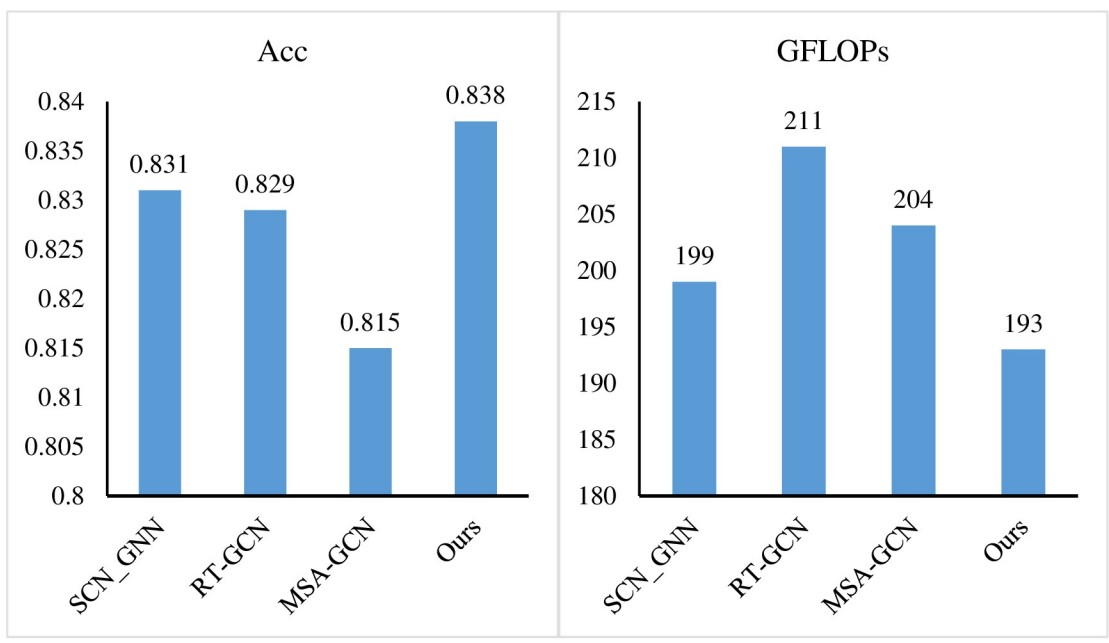

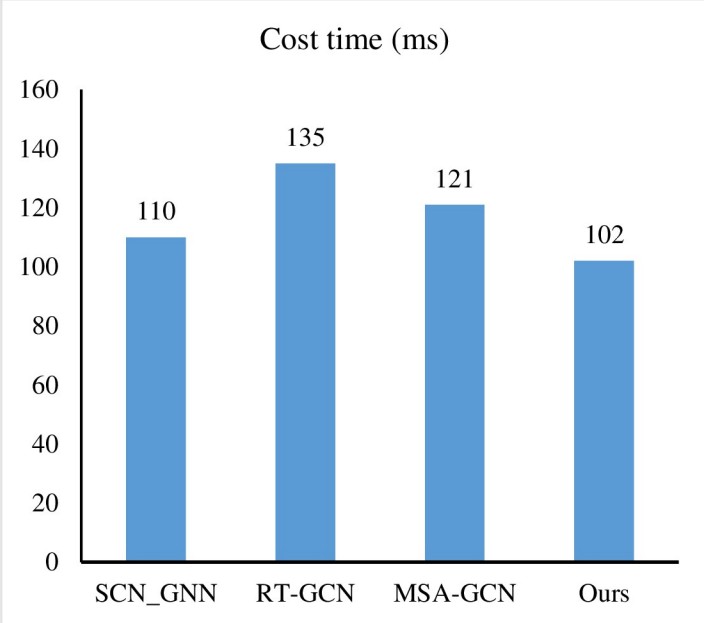

**Fig 8. The results of our MHN.**

stands out as a leader in the field of path planning due to its excellent performance in accuracy and global optimization. Its high performance demonstrated in practical applications will provide more reliable and efficient solutions for various path planning tasks.

## 4.4 Discussion

Through the above experiments of feature extraction method for adaptive multi-path selection, tourist personalized reasoning based on multi-layer hybrid network and path planning method based on hyper-network optimization, we fully demonstrate the effectiveness of path

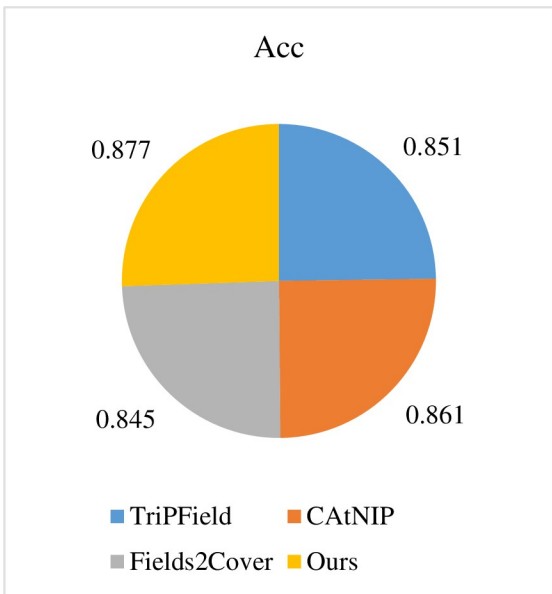
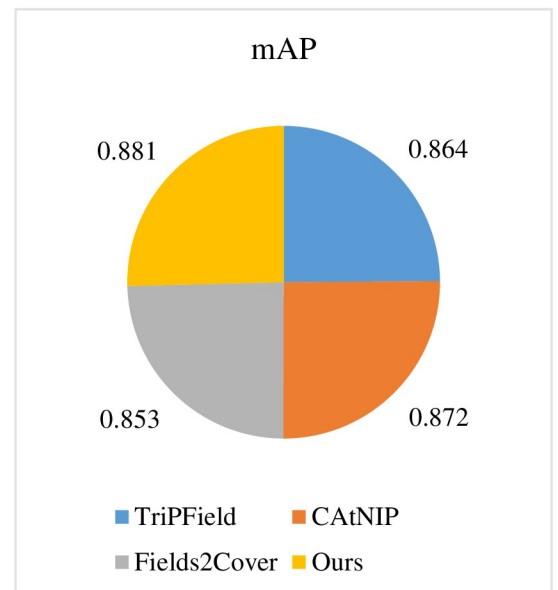

**Fig 9. Peformance comparison of HO with other methods.**

planning and itinerary customization of scenic spot based on multi-layer hybrid hyper-network optimization.

First, the feature extraction method of adaptive multi-path selection provides a more flexible and diversified feature representation for scenic spot path planning. By dynamically selecting and integrating different feature routes, the proposed method enables the model to more comprehensively and accurately capture various information such as scenic environment, tourist preferences and real-time changes. The experiments clearly demonstrate the adaptive feature extraction method has significant advantages in improving the accuracy and personalization of path planning. Second, the tourists' personalized reasoning based on multi-layer hybrid network fully demonstrates its excellent performance in the prediction of tourists' personalized demand by comparing with other models. The characteristics of high accuracy, low computational complexity and fast reasoning make the model more efficient in dealing with large-scale tourist data sets, and provide more intelligent and personalized services for scenic spots. Finally, the path planning approach by hypernetwork optimization provides a global optimization solution for the customization of scenic spots. Its excellent performance in accuracy and global optimization makes the generated path closer to the actual needs, and improves the quality of path planning and user satisfaction. Through the comparison with other path planning methods, we unequivocally affirm the notable benefits of the proposed approach in real-world applications.

This series of experiments proves the effectiveness of the path planning and itinerary customization of scenic spots based on multi-layer hybrid hypernetwork optimization. The multi-layer hybrid super-network optimization can fully utilize a variety of information, including tourists' personal preferences, historical behavior data, as well as geographical and cultural characteristics of the scenic spot, to customize an itinerary that better meets their individual needs. By optimizing the path planning, it can ensure that tourists' routes within the scenic area are both efficient and smooth. This not only saves tourists' time and energy, but also allows them to visit as many attractions within the scenic area as possible within a limited time. Additionally, it can be customized and adjusted according to different tourist needs and

scenic spot characteristics, adapting to various complex and changing scenarios. This not only provides a more intelligent and efficient solution for tourism business, but also provides useful enlightenment for tasks such as path planning and trip recommendation in other fields. The establishment of the intelligent service system of scenic spots based on deep learning provides strong support for the tourism science and technology in the future.

## 5. Conclusion

To boost the quality of scenic spots and provide tourists with the most desirable travel routes, we propose a path planning and itinerary customization method for scenic spots based on multi-layer hybrid hypernetwork optimization. Through the analysis of tourists' personalization, we successively propose an adaptive multi-path selection feature extraction method, a multi-layer hybrid network based personalized reasoning method for tourists, and a hypernetwork optimization path planning method. Thus, the personalized feature extraction of tourists is realized, the reasoning and acquisition of tourists' real needs are completed, and the best tour path for tourists is designed. Experiments show that our method can obtain an accuracy score of 0.877 and a mAP score of 0.881, which can provide a perfect solution for scenic spots to improve the quality of service.

## Acknowledgments

The author would like to thank the anonymous reviewers who have commented on this article.

## Author Contributions

**Conceptualization:** Chunqiao Song.

**Data curation:** Chunqiao Song.

**Formal analysis:** Chunqiao Song.

**Investigation:** Chunqiao Song.

**Methodology:** Chunqiao Song.

**Project administration:** Chunqiao Song.

**Validation:** Chunqiao Song.

**Writing – original draft:** Chunqiao Song.

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
