## [Decision Letter · Decision Letter 0]

19 Apr 2024

PONE-D-24-07129Scenic Spot Path Planning and Journey Customization Based on Multilayer Hybrid Hypernetwork OptimizationPLOS ONE

Dear Dr. Song,

Thank you for submitting your manuscript to PLOS ONE. After careful consideration, we feel that it has merit but does not fully meet PLOS ONE’s publication criteria as it currently stands. Therefore, we invite you to submit a revised version of the manuscript that addresses the points raised during the review process.

We look forward to receiving your revised manuscript.

Kind regards,

Yunhe Tong

Academic Editor

PLOS ONE

Journal Requirements:

4. Please upload a copy of Supporting Information Figure/Table/etc. S1 Fig.1-9 and S1 Table 1-4 which you refer to in your text on page 13.

Reviewers' comments:

Reviewer's Responses to Questions

**Comments to the Author**

1. Is the manuscript technically sound, and do the data support the conclusions?

Reviewer #1: Yes

Reviewer #2: No

2. Has the statistical analysis been performed appropriately and rigorously? 

Reviewer #1: N/A

Reviewer #2: I Don't Know

3. Have the authors made all data underlying the findings in their manuscript fully available?

Reviewer #1: Yes

Reviewer #2: Yes

4. Is the manuscript presented in an intelligible fashion and written in standard English?

Reviewer #1: Yes

Reviewer #2: No

5. Review Comments to the Author

Reviewer #1: This paper proposes scenic spot path planning and journey customization based on multilayer hybrid hypernetwork optimization.

There are still several aspects that need to be addressed.

1. The language expression appears to be overly redundant and requires simplification.

For example, the sentence "The research on path planning and itinerary customization of scenic spots involves many difficulties, which involve many aspects such as technology, data, and user behavior [6, 7]." may be "The research on path planning and itinerary customization of scenic spots involves many difficulties such as technology, data, and user behavior [6, 7]."

"First of all, ..." should be "First".

"Secondly, ..." should be "Second, ...".

"Finally, path planning and itinerary customization require... ..." should be "Therefore, path planning and itinerary customization require... ..."

2. The expressions of "multi-path" and "multi-way" should be consistent.

3. The abbreviation POI lacks its full name, e.g., Points of Interest.

4. The table for storing equations should have no borders.

5. The font of the variables in the formula is inconsistent with that in the main text.

6. In Equation (1), $Concat(\\phi_i)$ should be $Concat(\\phi_1, \\phi_2, \\dots, \\phi_i)$.

Moreover, the value range of $i$ is not clear and its setting in this paper is also unclear.

7. It's strange to have all the pictures attached, and the readability is also very low.

8. The equation (2) is difficult to be understood.

For example, the value range of $m$ and $n$ is not clear.

$O(i, j)$ and $I(i + m, j + n)$ appear indistinguishable but use different symbols.

9. The abbreviated form of "Adaptive multi-way selection Feature extraction (AMS)" is greatly difficult to understand.

10. It is difficult to find the support for "realize the personalized customization of tourists' itinerary" in the experiments.

Reviewer #2: Thank you for your submission to PLOS ONE. Your work focuses on an interesting topic: that of path planning with side-constraints. Unfortunately, though, the manuscript is organized in a manner that makes the problem unclear to most interested readers. Specifically, my comments follow.

1. The Introduction fails to provide the necessary background to understand the problem setup. Are we discussing about a shortest path problem with side-constraints? Is it supposed to be a tour or a circuit? What exactly is dynamic in the setup? Please consider updating this section with the necessary problem definition, perhaps followed by a visual representation of the problem.

2. The Literature review section is also very poorly written. At the moment, the literature review only mentions some of the works that have attempted to solve side-constrained network optimization problems. As an example, the most prominent method (that of Lagrangian optimization) for such problems is not even mentioned. Additionally, there is no synthesis in the Literature Review, as of now. A good literature review for a journal like PLOS ONE would require a better organization: at the moment, the literature review is "W solved the problem using Y, while X solved the problem using Z". Furthermore, the methodologies that you contrast your approach to in the Computational Results section are absent from the literature review. This section would need a thorough rewriting in order to become more concise and more inclusive.

3. The Methods section is missing important details for exactly how the model proposed is working. What makes it harder to follow is that some notation changes from subsection to subsection (see, for example, $D_i$ that is a feature in 3.1 but $D_i$ becomes a subset of scenic spots in 3.3). Traditionally, a full optimization problem would first be provided prior to tackling it using metaheuristics or AI-ML techniques.

4. While the data is available to download, a lot of details for the implementation and the selection of the hyperparameters is very unclear. The manuscript does not provide any reasoning for the selection of the hyperparameters and their impact of the metrics. One step that would help would be to make the code available to download and test, too.

5. The comparison to other methods is also unclear, as some of the methods compared to are optimizing for different objective functions.

In summary, I believe that there are significant edits to be made before the manuscript can be considered.

6. PLOS authors have the option to publish the peer review history of their article (what does this mean?). If published, this will include your full peer review and any attached files.

Reviewer #1: No

Reviewer #2: No

---

## [Author Response · Author response to Decision Letter 0]

5 May 2024

PONE-D-24-07129

Comments to the Author

1. Is the manuscript technically sound, and do the data support the conclusions?

Reviewer #1: Yes

Reviewer #2: No

2. Has the statistical analysis been performed appropriately and rigorously? 

Reviewer #1: N/A

Reviewer #2: I Don't Know

3. Have the authors made all data underlying the findings in their manuscript fully available?

Reviewer #1: Yes

Reviewer #2: Yes

4. Is the manuscript presented in an intelligible fashion and written in standard English?

Reviewer #1: Yes

Reviewer #2: No

5. Review Comments to the Author

Reviewer #1: This paper proposes scenic spot path planning and journey customization based on multilayer hybrid hypernetwork optimization.

There are still several aspects that need to be addressed.

1. The language expression appears to be overly redundant and requires simplification.

For example, the sentence "The research on path planning and itinerary customization of scenic spots involves many difficulties, which involve many aspects such as technology, data, and user behavior [6, 7]." may be "The research on path planning and itinerary customization of scenic spots involves many difficulties such as technology, data, and user behavior [6, 7]."

"First of all, ..." should be "First".

"Secondly, ..." should be "Second, ...".

"Finally, path planning and itinerary customization require... ..." should be "Therefore, path planning and itinerary customization require... ..."

Response: I check the sentences, as follows:

The endeavor of researching path planning and itinerary customization for scenic spots encompasses numerous challenges encompassing diverse aspects like technology, data, and user behavior [6,7].

Finally , the path planning and itinerary customization necessitates the development of intricate algorithms [11], which must take into account a myriad of variables and constraints.

2. The expressions of "multi-path" and "multi-way" should be consistent.

Response: I correct the expressions of "multi-path" and "multi-way", such as:

I adopt an adaptive multi-path selection feature extraction method.

Adaptive multi-path selection Feature extraction (AMS) is a technique used to capture diverse information and flexibly select feature extraction according to different input types.

3. The abbreviation POI lacks its full name, e.g., Points of Interest.

Response: I add the full name for these abbreviations, such as:

POI: points of interest.

GCN: Graph Convolution Network.

4. The table for storing equations should have no borders.

Response: I check the storing equations in tables, such as:

 (1)

 (2)

5. The font of the variables in the formula is inconsistent with that in the main text.

Response: I check the font in the main text, and correct them, such as:

where O(i,j) denotes the dot at (i,j). I(i+m,j+n) refers to the dot at (i+m, j+n) in the data. K(m, n) denotes the weight in the convolution kernel.

6. In Equation (1), $Concat(\\phi_i)$ should be $Concat(\\phi_1, \\phi_2, \\dots, \\phi_i)$.

Moreover, the value range of $i$ is not clear and its setting in this paper is also unclear.

Response: I correct the Equation (1) as follows:

 (1)

And, explain i as follows:

In the pre-training stage, for the feature Di of type i tourists where i∈R means the number of the entire types.

7. It's strange to have all the pictures attached, and the readability is also very low.

Response: I rewrite the introduction of the figures, such as:

The schematic representation of our approach is illustrated in Fig.1. First, to capture the behavioral characteristics of tourists more effectively, we have adopted an adaptive multi-path selection method for feature extraction. This method exhibits great flexibility and is able to intelligently select the appropriate feature extraction paths based on different types of information, such as users' historical itineraries and preferences for interesting points, thus more accurately capturing the personalized needs of tourists. Second, we have introduced personalized reasoning for tourists based on a multi-layer hybrid network. By deeply integrating tourist features at multiple levels, we can gain a more comprehensive understanding of their interests and preferences. This method provides more precise personalized information for subsequent route planning, contributing to the enhancement of personalized route planning. Finally, we have proposed a route planning method based on hypernetwork optimization, aiming to achieve efficient matching between tourists and scenic areas. By optimizing the hypernetwork structure of route planning, we can find the optimal path solutions while considering multiple factors. This helps to ensure that tourists have a better experience during their visit to the scenic area and improves the overall tourism efficiency. Overall, the route planning method proposed in this article, based on the optimization of a multi-layer hybrid hypernetwork, provides an innovative solution for improving the efficiency of route planning and the level of tourist experience by fully considering the needs of scenic areas and tourists, as well as introducing advanced technological means.

8. The equation (2) is difficult to be understood.

For example, the value range of $m$ and $n$ is not clear.

$O(i, j)$ and $I(i + m, j + n)$ appear indistinguishable but use different symbols.

Response: I add the explanation of equation (2) as follows:

and m, n denote the size of the kernel which are odd numbers. This formula represents that each output element in the convolution operation is the weighted sum of the input elements within the coverage of the convolutional kernel. The convolutional layer is able to extract local features from the data and pass these features to the next layer of the network for further processing.

9. The abbreviated form of "Adaptive multi-way selection Feature extraction (AMS)" is greatly difficult to understand.

Response: I adjust the abbreviation as follows:

Adaptive multi-path selection (AMS) feature extraction is a technique used to capture diverse information and flexibly select feature extraction according to different input types.

10. It is difficult to find the support for "realize the personalized customization of tourists' itinerary" in the experiments.

Response: I add the explanation about "realize the personalized customization of tourists' itinerary", as follows:

The multi-layer hybrid super-network optimization can fully utilize a variety of information, including tourists' personal preferences, historical behavior data, as well as geographical and cultural characteristics of the scenic spot, to customize an itinerary that better meets their individual needs. By optimizing the path planning, it can ensure that tourists' routes within the scenic area are both efficient and smooth. This not only saves tourists' time and energy, but also allows them to visit as many attractions within the scenic area as possible within a limited time. Additionally, it can be customized and adjusted according to different tourist needs and scenic spot characteristics, adapting to various complex and changing scenarios.

Reviewer #2: Thank you for your submission to PLOS ONE. Your work focuses on an interesting topic: that of path planning with side-constraints. Unfortunately, though, the manuscript is organized in a manner that makes the problem unclear to most interested readers. Specifically, my comments follow.

1. The Introduction fails to provide the necessary background to understand the problem setup. Are we discussing about a shortest path problem with side-constraints? Is it supposed to be a tour or a circuit? What exactly is dynamic in the setup? Please consider updating this section with the necessary problem definition, perhaps followed by a visual representation of the problem.

Response: I rewrite the introduction of our solving problem, as follows:

However, despite achieving certain success in scenic route planning and itinerary customization, these methods often overlook the personalized needs of tourists. The lack of personalized planning approaches makes it difficult to fully consider the tourists' diverse requirements. This may lead to tourists feeling uncomfortable or disappointed during their trips, and it also fails to fully leverage the unique features and advantages of the scenic spot, thereby reducing the overall quality of the tourism experience. To better meet the comprehensive needs of scenic route planning and itinerary customization, we need to further explore more intelligent methods. These methods should comprehensively consider factors such as tourists' personalized needs, the geographical and cultural characteristics of the scenic spot, as well as real-time environmental conditions, to create itineraries that not only meet tourists' expectations but also fully showcase the unique features of the scenic spot.

2. The Literature review section is also very poorly written. At the moment, the literature review only mentions some of the works that have attempted to solve side-constrained network optimization problems. As an example, the most prominent method (that of Lagrangian optimization) for such problems is not even mentioned. Additionally, there is no synthesis in the Literature Review, as of now. A good literature review for a journal like PLOS ONE would require a better organization: at the moment, the literature review is "W solved the problem using Y, while X solved the problem using Z". Furthermore, the methodologies that you contrast your approach to in the Computational Results section are absent from the literature review. This section would need a thorough rewriting in order to become more concise and more inclusive.

Response: I reorganize the Literature review section, as follows:

The current state of path planning research exhibits notable progress across various methodologies. Gupta et al. [22, 23] pioneered the integration of EMT into traditional evolutionary computation, introducing a multi-factor evolutionary algorithm. SyarifA et al. [24] introduced a fuzzy shortest path model, utilizing fuzzy numbers for evaluation and screening, and applied genetic algorithms for optimization. Wang et al. [26] enhanced genetic algorithms with a large-scale neighborhood search for optimal path solutions, showing superior performance compared to conventional genetic algorithms. Li et al. [27] proposed a novel genetic algorithm using continuous adjacent grid and unequal length chromosome coding for efficient robot obstacle avoidance in discrete grid environments. Fu et al. [28] designed a hybrid genetic algorithm integrating A* algorithm, demonstrating scalability and efficiency in route optimization problems. Guo et al. [29] developed a method for generating lane-level maps from sensor data, contributing to improved navigation systems. Ren et al. [30] mapped the search space into a deformable space, utilizing heuristic search for path planning, albeit with high computational complexity. Gonzalez et al. [31] employed a multi-resolution state lattice to enhance planning optimality and speed in trajectory search, particularly effective in complex global maps. Overall, these studies showcase advancements in path planning methodologies, from evolutionary algorithms to fuzzy logic and heuristic search, each contributing to improved efficiency and performance in various path planning scenarios.

In addition, Deep learning technology has significantly advanced path planning research, as evidenced by several notable contributions. Wang et al. [32] integrated deep learning with path planning techniques, employing a convolutional neural network (CNN) to guide the sampling procedure of the RRT* algorithm, resulting in improved path planning solutions. Q-Jeong et al. [33] utilized the triangle inequality to reduce path nodes in Q-RRT*, thereby decreasing path cost and accelerating convergence, although the sampling procedure remains random. Shiri et al. [34] introduced the oHJB algorithm, an online control algorithm for remote Unmanned Aerial Vehicles (UAVs) assisted by a neural network, enhancing UAV navigation capabilities. Tan et al. [35] proposed an AC-based neural network mission planning technique for heterogeneous multi-UAV systems, leveraging AC optimization to maximize benefits and minimize tasks. Yuan et al. [36] formulated UAV flight path tasks and employed CNN for object detection, coupled with optimization to identify optimal flight paths, enhancing navigation efficiency. Kim et al. [37] introduced a path planning approach utilizing a grid graph and optimization index to obtain a weighted network graph, simplifying route planning by transforming it into a minimum weight problem. Jabbar et al. [38] proposed a weighted network graph based on polygons, automatically generating shortest routes through an improved Dijkstra algorithm. However, this optimization method may generate multiple alternative routes and exhibit inefficiencies. These advancements underscore the integration of deep learning techniques into path planning, leading to more efficient and effective navigation solutions across various domains. 

Although there are numerous algorithms available to enhance path planning, when applied to practical multi-task scenarios, their iterative optimization processes often only handle a single path planning task. This limitation is particularly evident in multi-task environments, which typically involve multiple target points, dynamic obstacles, and real-time map information. Most existing path planning algorithms are based on the single-task assumption, meaning they tackle only one path planning task from start to finish at a time. However, in complex multi-task scenarios, multiple path planning tasks need to be considered simultaneously, and these tasks may have mutual interferences and constraints. Therefore, this paper mainly explores how to effectively represent and handle the complex constraints and interferences in multi-task environments.

3. The Methods section is missing important details for exactly how the model proposed is working. What makes it harder to follow is that some notation changes from subsection to subsection (see, for example, $D_i$ that is a feature in 3.1 but $D_i$ becomes a subset of scenic spots in 3.3). Traditionally, a full optimization problem would first be provided prior to tackling it using metaheuristics or AI-ML techniques.

Response: I rewrite the overall introduction of our model, as follows:

The schematic representation of our approach is illustrated in Fig.1. First, to capture the behavioral characteristics of tourists more effectively, we have adopted an adaptive multi-path selection method for feature extraction. This method exhibits great flexib

---

## [Decision Letter · Decision Letter 1]

26 Jun 2024

PONE-D-24-07129R1Scenic Spot Path Planning and Journey Customization Based on Multilayer Hybrid Hypernetwork OptimizationPLOS ONE

Dear Dr. Song,

Thank you for submitting your manuscript to PLOS ONE. After careful consideration, we feel that it has merit but does not fully meet PLOS ONE’s publication criteria as it currently stands. Therefore, we invite you to submit a revised version of the manuscript that addresses the points raised during the review process.

We look forward to receiving your revised manuscript.

Kind regards,

Yunhe Tong

Academic Editor

PLOS ONE

Journal Requirements:

Additional Editor Comments:

Please see below for the comments:

To realize the efficient matching between tourists and scenic spots, this paper proposes (1) the feature extraction method of adaptive multi-path selection (AMS), (2) the tourist personalized reasoning method based on multi-layer hybrid network (MHN), and (3) the path planning method based on hyper-network optimization (HO).

However, there are still several aspects need be improvemed.

(1) The notations are chaotic and their connections are weak.

(2) The optimization objective function (see, Eq. (5)) is uninformative and very difficult to understand.

(3) The accuracy of expression urgently needs to be improved. For example, the ``OH'' listed in Table 2 should be ``HO''.

(4) The best result should be highlighted, such as in bold.

(5) If the AMS, MHN, and HO are not considered, what exactly is the baseline-model?

(6) In the 20th page of this PDF file, there is a mistaken code between 0.50 and 0.95.

(7) The expression of first person such as ``We'' or ``I'' should be unified.

(8) The path planning and path prediction are two completely different problems.

Reviewers' comments:

Reviewer's Responses to Questions

**Comments to the Author**

1. If the authors have adequately addressed your comments raised in a previous round of review and you feel that this manuscript is now acceptable for publication, you may indicate that here to bypass the “Comments to the Author” section, enter your conflict of interest statement in the “Confidential to Editor” section, and submit your "Accept" recommendation.

Reviewer #1: All comments have been addressed

2. Is the manuscript technically sound, and do the data support the conclusions?

Reviewer #1: Yes

3. Has the statistical analysis been performed appropriately and rigorously? 

Reviewer #1: Yes

4. Have the authors made all data underlying the findings in their manuscript fully available?

Reviewer #1: Yes

5. Is the manuscript presented in an intelligible fashion and written in standard English?

Reviewer #1: Yes

6. Review Comments to the Author

Reviewer #1: To realize the efficient matching between tourists and scenic spots, this paper proposes (1) the feature extraction method of adaptive multi-path selection (AMS), (2) the tourist personalized reasoning method based on multi-layer hybrid network (MHN), and (3) the path planning method based on hyper-network optimization (HO).

However, there are still several aspects need be improvemed.

(1) The notations are chaotic and their connections are weak.

(2) The optimization objective function (see, Eq. (5)) is uninformative and very difficult to understand.

(3) The accuracy of expression urgently needs to be improved. For example, the ``OH'' listed in Table 2 should be ``HO''.

(4) The best result should be highlighted, such as in bold.

(5) If the AMS, MHN, and HO are not considered, what exactly is the baseline-model?

(6) In the 20th page of this PDF file, there is a mistaken code between 0.50 and 0.95.

(7) The expression of first person such as ``We'' or ``I'' should be unified.

(8) The path planning and path prediction are two completely different problems.

7. PLOS authors have the option to publish the peer review history of their article (what does this mean?). If published, this will include your full peer review and any attached files.

Reviewer #1: No

---

## [Author Response · Author response to Decision Letter 1]

1 Jul 2024

PONE-D-24-07129R1

Comments to the Author

6. Review Comments to the Author

Reviewer #1: To realize the efficient matching between tourists and scenic spots, this paper proposes (1) the feature extraction method of adaptive multi-path selection (AMS), (2) the tourist personalized reasoning method based on multi-layer hybrid network (MHN), and (3) the path planning method based on hyper-network optimization (HO).

However, there are still several aspects need be improved.

(1)The notations are chaotic and their connections are weak.

Response: I modify the notations as follows:

 (3)

 (5)

(2)The optimization objective function (see, Eq. (5)) is uninformative and very difficult to understand.

Response:I add the content to understand the optimization objective function, as follows:

When considering how to plan a tour route to maximize tourist satisfaction, equation 5 can serve as a crucial mathematical modeling tool to guide our decision-making. In the tourism industry, the design of a route that maximizes tourist satisfaction takes into account factors such as transportation, accommodation, catering, attractions, and tourists' time.

(3)The accuracy of expression urgently needs to be improved. For example, the ``OH'' listed in Table 2 should be ``HO''.

Response: I correct the mistake in Table 2, as follows:

Table 2 Ablation experiments

AMS MHN HO mAP Acc L T (ms)

 0.775 0.782 22 234

O 0.811 0.804 18 192

 O 0.804 0.811 17 189

 O 0.816 0.807 19 199

O O 0.834 0.838 15 201

O O 0.842 0.842 14 215

 O O 0.839 0.833 15 228

O O O 0.881 0.877 12 236

(4)The best result should be highlighted, such as in bold.

Response: I highlight the best results in tables, as follows:

Table 3 Comparison AMS with other methods

Methods mAP mAP50 mAP75 mAP95

CNN-FPN 0.753 0.789 0.762 0.732

Self-Attention 0.797 0.835 0.803 0.753

Uniformer 0.801 0.824 0.811 0.786

LiConvFormer 0.785 0.831 0.796 0.767

Ours 0.811 0.846 0.814 0.796

Table 4 Compare KPD with other methods

Methods Acc GFLOPs Cost time (ms)

SCN_GNN 0.831 199 110

RT-GCN 0.829 211 135

MSA-GCN 0.815 204 121

Ours 0.838 193 102

(5)If the AMS, MHN, and HO are not considered, what exactly is the baseline-model?

Response: I introduce the baseline in Section 4.2, as follows:

We employ the CNN as our baseline. Combining the results of attraction recommendations and geographic information data, CNN can assist in planning optimal travel routes. By analyzing information such as the geographic location, traffic conditions, and tourist flow of attractions, CNN can help tourists select the most suitable tour sequence and transportation mode.

(6)In the 20th page of this PDF file, there is a mistaken code between 0.50 and 0.95.

Response: I correct this mistake, as follows:

We used mAP50, mAP75, mAP95, and mAP to represent the mean average precision (AP) of all classes when the Intersection over Union (IoU) was is equal to 0.50, 0.75, 0.95, and 0.50 : 0.95, respectively.

(7)The expression of first person such as ``We'' or ``I'' should be unified.

Response: I unify the ``We'' or ``I'', as follows:

Finally, we conduct a thorough investigation into the performance of the hyper-network optimization-based path planning method (HO), and we utilize a nonlinear objective function for all comparisons.

(8)The path planning and path prediction are two completely different problems.

Response:

Response: I add the introduction of path planning as follows:

Tourism route planning research involves multiple aspects, including understanding the destination, developing the itinerary theme, designing diversified travel routes, arranging a reasonable schedule, selecting transportation modes, arranging accommodations, planning meals, and organizing activities.

---

## [Editor Report · Decision Letter 2]

3 Jul 2024

PONE-D-24-07129R2Scenic Spot Path Planning and Journey Customization Based on Multilayer Hybrid Hypernetwork OptimizationPLOS ONE

Dear Dr. Song,

Thank you for submitting your manuscript to PLOS ONE. After careful consideration, we feel that it has merit but does not fully meet PLOS ONE’s publication criteria as it currently stands. Therefore, we invite you to submit a revised version of the manuscript that addresses the points raised during the review process. The reviewer only suggested minor revisions. Please see below for detailed comments.

We look forward to receiving your revised manuscript.

Kind regards,

Yunhe Tong

Academic Editor

PLOS ONE

Journal Requirements:

Reviewers' comments:

To realize the efficient matching between tourists and scenic spots, this paper proposes (1) the feature extraction method of adaptive multi-path selection (AMS), (2) the tourist personalized reasoning method based on multi-layer hybrid network (MHN), and (3) the path planning method based on hyper-network optimization (HO).

However, there are still several aspects need be improvemed.

(1) The notations are chaotic and their connections are weak.

(2) The optimization objective function (see, Eq. (5)) is uninformative and very difficult to understand.

(3) The accuracy of expression urgently needs to be improved. For example, the ``OH'' listed in Table 2 should be ``HO''.

(4) The best result should be highlighted, such as in bold.

(5) If the AMS, MHN, and HO are not considered, what exactly is the baseline-model?

(6) In the 20th page of this PDF file, there is a mistaken code between 0.50 and 0.95.

(7) The expression of first person such as ``We'' or ``I'' should be unified.

(8) The path planning and path prediction are two completely different problems.

---

## [Author Response · Author response to Decision Letter 2]

15 Jul 2024

PONE-D-24-07129R1

Comments to the Author

6. Review Comments to the Author

Reviewer #1: To realize the efficient matching between tourists and scenic spots, this paper proposes (1) the feature extraction method of adaptive multi-path selection (AMS), (2) the tourist personalized reasoning method based on multi-layer hybrid network (MHN), and (3) the path planning method based on hyper-network optimization (HO).

However, there are still several aspects need be improved.

(1)The notations are chaotic and their connections are weak.

Response: I modify the notations as follows:

 (3)

 (5)

(2)The optimization objective function (see, Eq. (5)) is uninformative and very difficult to understand.

Response:I add the content to understand the optimization objective function, as follows:

When considering how to plan a tour route to maximize tourist satisfaction, equation 5 can serve as a crucial mathematical modeling tool to guide our decision-making. In the tourism industry, the design of a route that maximizes tourist satisfaction takes into account factors such as transportation, accommodation, catering, attractions, and tourists' time.

(3)The accuracy of expression urgently needs to be improved. For example, the ``OH'' listed in Table 2 should be ``HO''.

Response: I correct the mistake in Table 2, as follows:

Table 2 Ablation experiments

AMS MHN HO mAP Acc L T (ms)

 0.775 0.782 22 234

O 0.811 0.804 18 192

 O 0.804 0.811 17 189

 O 0.816 0.807 19 199

O O 0.834 0.838 15 201

O O 0.842 0.842 14 215

 O O 0.839 0.833 15 228

O O O 0.881 0.877 12 236

(4)The best result should be highlighted, such as in bold.

Response: I highlight the best results in tables, as follows:

Table 3 Comparison AMS with other methods

Methods mAP mAP50 mAP75 mAP95

CNN-FPN 0.753 0.789 0.762 0.732

Self-Attention 0.797 0.835 0.803 0.753

Uniformer 0.801 0.824 0.811 0.786

LiConvFormer 0.785 0.831 0.796 0.767

Ours 0.811 0.846 0.814 0.796

Table 4 Compare KPD with other methods

Methods Acc GFLOPs Cost time (ms)

SCN_GNN 0.831 199 110

RT-GCN 0.829 211 135

MSA-GCN 0.815 204 121

Ours 0.838 193 102

(5)If the AMS, MHN, and HO are not considered, what exactly is the baseline-model?

Response: I introduce the baseline in Section 4.2, as follows:

We employ the CNN as our baseline. Combining the results of attraction recommendations and geographic information data, CNN can assist in planning optimal travel routes. By analyzing information such as the geographic location, traffic conditions, and tourist flow of attractions, CNN can help tourists select the most suitable tour sequence and transportation mode.

(6)In the 20th page of this PDF file, there is a mistaken code between 0.50 and 0.95.

Response: I correct this mistake, as follows:

We used mAP50, mAP75, mAP95, and mAP to represent the mean average precision (AP) of all classes when the Intersection over Union (IoU) was is equal to 0.50, 0.75, 0.95, and 0.50 : 0.95, respectively.

(7)The expression of first person such as ``We'' or ``I'' should be unified.

Response: I unify the ``We'' or ``I'', as follows:

Finally, we conduct a thorough investigation into the performance of the hyper-network optimization-based path planning method (HO), and we utilize a nonlinear objective function for all comparisons.

(8)The path planning and path prediction are two completely different problems.

Response:

Response: I add the introduction of path planning as follows:

Tourism route planning research involves multiple aspects, including understanding the destination, developing the itinerary theme, designing diversified travel routes, arranging a reasonable schedule, selecting transportation modes, arranging accommodations, planning meals, and organizing activities.

---

## [Editor Report · Decision Letter 3]

18 Jul 2024

Scenic Spot Path Planning and Journey Customization Based on Multilayer Hybrid Hypernetwork Optimization

PONE-D-24-07129R3

Dear Dr. Song,

We’re pleased to inform you that your manuscript has been judged scientifically suitable for publication and will be formally accepted for publication once it meets all outstanding technical requirements.

Kind regards,

Yunhe Tong

Academic Editor

PLOS ONE
---

## [Editor Report · Acceptance letter]

3 Oct 2024

PONE-D-24-07129R3 

PLOS ONE

Dear Dr. Song, 

I'm pleased to inform you that your manuscript has been deemed suitable for publication in PLOS ONE. Congratulations! Your manuscript is now being handed over to our production team.

Kind regards, 

on behalf of

Dr. Yunhe Tong 

Academic Editor

PLOS ONE